# Immunogenicity of SARS-CoV-2 Vaccination Schedules Including a Booster Dose in Patients with Systemic Lupus Erythematosus: Data from a Prospective Multicenter Study

**DOI:** 10.3390/vaccines13020127

**Published:** 2025-01-27

**Authors:** Natália Sarzi Sartori, Ketty Lysie Libardi Lira Machado, Samira Tatiyama Miyamoto, Flávia Zon Pretti, Maria da Penha Gomes Gouveia, Yasmin Gurtler Pinheiro de Oliveira, Vanezia Gonçalves da Silva, Filipe Faé, Ana Paula Neves Burian, Karina Rosemarie Lallemand Tapia, Anna Carolina Simões Moulin, Luiza Lorenzoni Grillo, Paula dos Santos Athayde, Helena da Silva Corona, Sabrina de Souza Ramos, Flávia Maria Matos Melo Campos Peixoto, Priscila Dias Cardoso Ribeiro, Vanessa de Oliveira Magalhães, Mariana Freitas de Aguiar, Erika Biegelmeyer, Cristiane Kayser, Alexandre Wagner Silva de Souza, Charlles Heldan de Moura Castro, Juliana Bühring, Sandra Lúcia Euzébio Ribeiro, Sérgio Henrique Oliveira dos Santos, Clara Pinheiro Martins, Jonathan Willian da Silva Rodrigues, Marcos Mavignier Sousa Dias, Bruna Guimarães Dutra, Camila Maria Paiva França Telles, Samuel Elias Basualto Dias, Rodrigo Poubel Vieira de Rezende, Katia Lino Baptista, Rodrigo Cutrim Gaudio, Ana Karla Guedes de Melo, Valéria Bezerra da Silva, Vitor Alves Cruz, Jozelia Rêgo, Rejane Maria Rodrigues de Abreu Vieira, Adah Sophia Rodrigues Vieira, Adriana Maria Kakehasi, Anna Carolina Faria Moreira Gomes Tavares, Victória Dornelas Paz Carvalho, Renata Henriques de Azevedo, Valderilio Feijó Azevedo, Olindo Assis Martins-Filho, Vanessa Peruhype-Magalhães, Andrese Aline Gasparin, Vanessa Hax, Valéria Valim, Gilda Aparecida Ferreira, Andréa Teixeira-Carvalho, Edgard Torres dos Reis-Neto, Emília Inoue Sato, Marcelo de Medeiros Pinheiro, Viviane Angelina de Souza, Ricardo Machado Xavier, Gecilmara Salviato Pileggi, Odirlei André Monticielo

**Affiliations:** 1Serviço de Reumatologia, Hospital de Clínicas de Porto Alegre, Universidade Federal do Rio Grande do Sul (UFRGS), Porto Alegre 90010-150, RS, Brazil; andresegasparin@gmail.com (A.A.G.); vanessahax@gmail.com (V.H.); rxavier10@gmail.com (R.M.X.); 2Hospital Universitário Cassiano Antônio Moraes (HUCAM), Universidade Federal do Espírito Santo (UFES), Vitória 29041-295, ES, Brazil; drakettymachado@gmail.com (K.L.L.L.M.); sa.miyamoto@hotmail.com (S.T.M.); flaviazon@hotmail.com (F.Z.P.); mpgomesgov@gmail.com (M.d.P.G.G.); yasmingurtler@hotmail.com (Y.G.P.d.O.); vanezia.hucam@gmail.com (V.G.d.S.); fae.filipe@gmail.com (F.F.); karinalallemand@gmail.com (K.R.L.T.); annasmoulin@gmail.com (A.C.S.M.); luizalgrillo@gmail.com (L.L.G.); paula.athaydeifes@gmail.com (P.d.S.A.); helena.corona@edu.ufes.br (H.d.S.C.); sasouzaramos@gmail.com (S.d.S.R.); val.valim@gmail.com (V.V.); 3Centro de Referências para Imunobiológicos Especiais (CRIE) da Secretaria de Saúde do Estado do Espírito Santo, Vitória 29050-710, ES, Brazil; anapaulaburian@gmail.com; 4Escola Paulista de Medicina (EPM), Universidade Federal de São Paulo (UNIFESP), São Paulo 04023-062, SP, Brazil; flaviacampospeixoto@gmail.com (F.M.M.M.C.P.); pri.dcr@gmail.com (P.D.C.R.); vanessa.reumato@gmail.com (V.d.O.M.); mari.f.aguiar@hotmail.com (M.F.d.A.); erika.biegel@gmail.com (E.B.); cristiane.kayser@unifesp.br (C.K.); alexandre_wagner@uol.com.br (A.W.S.d.S.); cheldan@uol.com.br (C.H.d.M.C.); edgard.torres@unifesp.br (E.T.d.R.-N.); eisato@unifesp.br (E.I.S.); mpinheiro@uol.com.br (M.d.M.P.); gecilmara@gmail.com (G.S.P.); 5Department of Rheumatology, Universidade Federal do Amazonas (UFAM), Manaus 69080-900, AM, Brazil; jubuhring@hotmail.com (J.B.); sandraler04@gmail.com (S.L.E.R.); sergio.henrique@live.com (S.H.O.d.S.); mp.clara9@gmail.com (C.P.M.); jonathanufam@gmail.com (J.W.d.S.R.); mavignierdias@gmail.com (M.M.S.D.); brunagdutraa@gmail.com (B.G.D.); camilapaiva2003@hotmail.com (C.M.P.F.T.); samuel.ebd@gmail.com (S.E.B.D.); 6Department of Rheumatology, Universidade Federal Fluminense (UFF), Niterói 24220-900, RJ, Brazil; ropoubel@id.uff.br (R.P.V.d.R.); linokatia@gmail.com (K.L.B.); rodrigogaudio@yahoo.com.br (R.C.G.); 7Hospital Universitário Lauro Wanderley, Universidade Federal da Paraíba (UFPB), João Pessoa 58050-585, PB, Brazil; anakarlagmelo@gmail.com (A.K.G.d.M.); valeria.silvace@gmail.com (V.B.d.S.); 8Department of Rheumatology, Universidade Federal de Goiás (UFG), Goiânia 74690-900, GO, Brazil; vitorcruz@ufg.br (V.A.C.); jobranca2007@gmail.com (J.R.); 9Department of Rheumatology, Universidade de Fortaleza (UNIFOR), Fortaleza 60811-905, CE, Brazil; rejaneavieira@gmail.com (R.M.R.d.A.V.); adahsophia0820@gmail.com (A.S.R.V.); 10Hospital das Clínicas da Universidade Federal de Minas Gerais, Belo Horizonte 31270-901, MG, Brazil; amkakehasi@gmail.com (A.M.K.); annacgtavares@gmail.com (A.C.F.M.G.T.); 11Department of Rheumatology, Universidade Federal de Juiz de Fora (UFJF), Juiz de Fora 36036-900, MG, Brazil; victoriadornelaspc@gmail.com (V.D.P.C.); renata.reumato@gmail.com (R.H.d.A.); 12Edumed—Educação em Saúde S/S Ltd., Curitiba 80440-080, PR, Brazil; valderilio@hotmail.com; 13Instituto Renè Rachou, Fundação Oswaldo Cruz (FIOCRUZ-Minas), Belo Horizonte 30190-002, MG, Brazil; olindo.filho@fiocruz.br (O.A.M.-F.); vanessa.pascoal@fiocruz.br (V.P.-M.); andrea.teixeira@fiocruz.br (A.T.-C.); 14Locomotor System Department, Faculdade de Medicina, Universidade Federal de Minas Gerais (UFMG), Belo Horizonte 30130-100, MG, Brazil; gildaferreira9@gmail.com; 15Faculdade de Medicina, Universidade Federal de Juiz de Fora, Juiz de Fora 36036-900, MG, Brazil; vivi.reumato@gmail.com

**Keywords:** systemic lupus erythematosus, immunogenicity, SARS-CoV-2 vaccines, COVID-19 vaccines

## Abstract

Objective: To evaluate the humoral response to and impact of SARS-CoV-2 vaccination in patients with systemic lupus erythematosus in a multicenter cohort design. Methods: Data for this analysis were obtained from the Study of Safety, Effectiveness and Duration of Immunity after Vaccination against SARS-CoV-2 in Patients with Immune-Mediated Inflammatory Diseases (SAFER), a prospective, multicenter, phase IV, real-world study conducted across different regions of Brazil from June/2021 to March/2024. Patients aged >18 years with systemic lupus erythematosus (SLE) who received any one of the SARS-CoV-2 vaccines approved by the Brazilian health regulatory agency (CoronaVac [inactivated SARS-CoV-2 vaccine], ChAdOx-1 [AstraZeneca], or BNT162b2 [Pfizer-BioNTech]) were included. Immunogenicity was assessed in pre- and post-vaccination blood samples, and patients were monitored in person and remotely for the occurrence and severity of COVID-19. Results: Two hundred and thirty-five patients with SLE who had completed their vaccination schedules (two doses + booster dose) were included in this study. Most patients were female (89.3%) and had low disease activity or were in remission (72.4%); the majority were also on some form of immunosuppressive therapy (58.1%). One hundred and sixteen patients received two doses of CoronaVac followed by one dose of BNT162b2 (Pfizer-BioNTech) vaccine, eighty-seven received two doses of ChAdOx1-S (AstraZeneca) followed by one dose of BNT162b2 (Pfizer-BioNTech) vaccine, and thirty-two received three doses of BNT162b2 (Pfizer-BioNTech) vaccine. Twenty-eight cases of COVID-19, none meeting criteria for severe COVID-19, were recorded in patients with respiratory symptoms after the second dose of a SARS-CoV-2 vaccine. Regarding immunogenicity, an increase in seroconversion rate was observed following consecutive vaccine doses, with no difference between vaccination schedules, reaching 97.57% seropositivity after a booster dose. The geometric mean IgG titers differed between the different vaccination schedules after the first and the second vaccine dose, being lowest for the CoronaVac-based schedule, but titers were similar after the administration of a booster dose. Conclusion: In patients with SLE, SARS-CoV-2 vaccines are immunogenic, inducing a robust humoral response. No severe outcomes associated with death or hospitalization were found in the evaluated patient sample. Complete vaccination schedules including a booster dose induced higher humoral responses than incomplete schedules, especially in patients initially immunized with an inactivated virus vaccine schedule and those with a suboptimal humoral response.

## 1. Introduction

Systemic lupus erythematosus (SLE) is associated with abnormalities in the humoral and cellular immune responses. These changes, associated both with the immunosuppressive therapy needed to control disease manifestations and with active disease itself, are predisposing factors for greater susceptibility to infections and progression to serious outcomes of such infections [1,2]. Several previous studies have shown that infections are the leading cause of both early and late mortality in patients with SLE [3,4].

Individuals with SLE have an approximately sixfold risk of serious infections compared to the general population [5]. Since the emergence of COVID-19, the infection caused by the SARS-CoV-2 virus has been a cause of great concern for this vulnerable population. Among immune-mediated rheumatic diseases, SLE is associated with some of the most severe manifestations of SARS-CoV-2 infection and some of the highest hospitalization rates for COVID-19 [6,7]. Data from previous studies during pre-vaccination waves suggest hospitalization rates of around 20%, with one reporting a rate of over 50% [8,9].

Immunization is one of the most effective tools for preventing infections as a public health strategy, contributing to a reduced incidence of serious cases of infectious diseases and, consequently, reducing interpersonal spread [10]. Patients with immune-mediated rheumatic diseases exhibit different degrees of immunosuppression depending on their therapeutic regimen, their level of disease activity, and the manifestations of their disease [11]. Live-attenuated vaccines are generally contraindicated in this patient population due to the risk of the small amount of live virus particles present inducing uncontrolled infection, but they may be considered in selected patients with a lower degree of immunosuppression and a favorable risk–benefit ratio. Conversely, inactivated vaccines are recommended for use in immunosuppressed patients, preferably before starting immunosuppressive therapy [12,13].

Data on vaccine response in SLE are controversial. The immune response to vaccines and their real-world effectiveness are affected by several host factors. Therefore, immunogenicity is hypothesized to be lower in patients with SLE. A previous meta-analysis that evaluated the efficacy of the influenza vaccine demonstrated lower immunogenicity in individuals with SLE when compared to healthy controls, although the level of immunity achieved was still considered protective [14].

Concerns regarding vaccine efficacy in these patients were amplified for SARS-CoV-2 vaccines, which underwent a fast-tracked emergency marketing authorization process and were developed on a wide range of different platforms. Given these uncertainties, the objective of this study was to evaluate the magnitude of the humoral response and the real-world immunogenicity of these vaccines in individuals with SLE.

## 2. Materials and Methods

### 2.1. Study Design and Population

This study evaluated patients with SLE included in the multicenter Study of Safety, Effectiveness and Duration of Immunity after Vaccination against SARS-CoV-2 in Patients with Immune-Mediated Inflammatory Diseases (SAFER), a Brazilian observational, prospective, phase IV cohort study started in June/2021 and completed in March/2024. Patients were enrolled from June/2021 to September/2021 and followed for 4 weeks post receiving the third vaccine dose. Patients over 18 years of age who met the 2019 American College of Rheumatology/European Alliance of Rheumatology Associations (ACR/EULAR) classification criteria for SLE [15] and who had received any SARS-CoV-2 vaccine as recommended in the Brazilian National Immunization Plan were included. The exclusion criteria were history of previous adverse vaccine reaction, pregnancy, and immunosuppression for any other reason (HIV, organ transplantation, malignancy).

The included patients received a complete vaccination schedule (2-dose regimen plus booster) against SARS-CoV-2 with vaccines approved by the Brazilian Health Surveillance Agency, namely CoronaVac (inactivated SARS-CoV-2 vaccine), ChAdOx-1 (AstraZeneca, Cambridge, UK), and BNT162b2 (Pfizer-BioNTech, Mainz, Germany). All vaccine doses were administered as indicated by a medical professional under supervision.

Patients were evaluated in person at different time points before and after vaccine exposure (baseline, T1; before the 2nd and 3rd dose, T2 and T3, respectively; four weeks after the 3rd dose, T4; twelve weeks after the 3rd dose, T5; twenty-four weeks after the 3rd dose, T6; and 1 year after the 3rd dose, T7), and by telephone monitoring conducted in the intervals between in-person visits (Figure 1).

### 2.2. Variables of Interest

Demographic data (age, sex, and race), comorbidities, and history of COVID-19 infection prior to vaccination were recorded at the baseline visit. Disease activity score (SLEDAI-2K)—categorized into remission (SLEDAI-2K = 0), low activity (SLEDAI-2K 1–5), or moderate-to-high activity (SLEDAI 2K > 6)—and degree of immunosuppression were recorded at the baseline and subsequent assessments. The degree of immunosuppression was assessed following the recommendations of the Brazilian Society of Rheumatology, consistent with the risk of infectious events conferred by each medication (Box 1) [16].

Box 1Position of the Brazilian Society of Rheumatology (SBR) regarding the degree of immunosuppressants of immunobiologicals.Patients considered not immunosuppressed  no drugs  use of sulfasalazine or hydroxychloroquine  using topical, inhaled, intra-articular corticosteroidsPatients considered to be under low degree of immunosuppression  methotrexate ≤ 20 mg/week  leflunomide dose of 20 g dailyPatients considered to be under a high degree of immunosuppression  daily corticosteroids in doses ≥ 10 mg/day of prednisone or equivalent for more than 14 days  pulse therapy with methylprednisolone  mycophenolic acid, cyclosporine, tacrolimus, cyclophosphamide, azathioprine  JAK inhibitors (small molecules)  immunobiologicals

Biological specimens were collected at all visits to measure the serologic response to vaccination, assessed by chemiluminescence methods. The Elecsys^®^ Anti-SARS-CoV-2 S immunoassay (Roche, Basel, Switzerland), validated by the World Health Organization, was used. A cut-off index (COI) ≥ 1.0 U/mL was defined as positive [17].

The surveillance of symptomatic COVID-19 cases was carried out remotely, periodically (every 2 weeks), or reactively. Patients who developed symptoms consistent with COVID-19 for up to 1 month after their last vaccine dose were advised to undergo nasal swab collection for PCR testing; the outcomes were monitored by the study team.

The date of vaccine administration and type of vaccine administered were recorded and categorized by the vaccine platform of the first 2 doses. For cases of COVID-19, clinical presentation, date of symptom onset, and duration and severity of infection were assessed.

Data from all participating centers were entered into a unified electronic platform (REDCap—Research Electronic Data Capture, https://redcap.reumatologia.org.br/, accessed on 15 November 2024).

### 2.3. Statistical Analysis

Analyses were performed using Stata (v.17) and R (v.4.2.0) software. For all tests, statistical significance was accepted at the 5% level and 95% confidence intervals were calculated.

We performed a descriptive analysis of demographic data, comorbidities, disease activity score, and degree of immunosuppression (using the definition recommended by the Brazilian Society of Rheumatology), stratified by vaccine platform. For categorical variables, proportions between groups were compared using the chi-square and Fisher’s exact tests. For continuous variables, proportions between groups were expressed as means and standard deviations or medians and interquartile ranges, as appropriate. We analyzed these variables using ANOVA and the Wilcoxon test (2 groups) or Kruskal–Wallis test (>2 groups), respectively.

Humoral immunogenicity data were evaluated as the seroconversion rate by vaccine group, according to collection time and treatment type. Within each group, we compared the proportions of seroconversion as well as the geometric means of the antibody titers. For the analysis of IgG titers, data were normalized by log_10_-transformation. The analysis of normalized IgG titers over time was then performed using the nonparametric Wilcoxon/Mann–Whitney test with Bonferroni correction. A multivariate (linear) regression model for IgG titers was adjusted for disease activity and degree of immunosuppression.

### 2.4. Ethical Aspects

The study was submitted to the National Research Ethics Commission for approval (CAAE 43479221.0.1001.5505) and to the Research Ethics Committees of all participating centers, and was conducted in accordance with the applicable guidelines and standards that regulate research on human subjects. All participants signed an Informed Consent Form (ICF) after being informed of the objective and protocol of the study.

All biosafety guidelines and Good Clinical Laboratory Practices were followed.

## 3. Results

The present study included a total of 445 patients with SLE, of whom 235 patients over 18 years of age completed a three-dose COVID-19 vaccination schedule (Figure 2).

Of these, 210 (89.3%) were women and 25 (10.6%) were men, with an average age of 38 years. All individuals were Brazilian. Regarding ethnicity/skin color, 109 (46.3%) self-identified as brown and 87 (37%) as white. The median of disease duration was 10 years (interquartile range 5–16 years). In this group, 39.1% of individuals had no other comorbidities. Among patients reporting comorbidities, 27.2% had hypertension and 11.4% had obesity; other comorbidities were hypothyroidism, osteonecrosis, and dyslipidemia. According to the SLEDAI-2K score, most patients (72.4%) were in remission or had low disease activity. Regarding the severity of immunosuppression, 135 (58.1%) had a high degree of immunosuppression and 70 (30.1%) were not immunosuppressed [Table 1]. Concerning pharmacotherapy, 82.5% of patients were on hydroxychloroquine; among immunosuppressants, azathioprine and mycophenolate were equally common (22.98%). Approximately 50% of patients were on oral glucocorticoids, most (48.2%) at doses of up to 5 mg per day (Table 2).

Regarding vaccine platforms and immunization schedules, 116 patients received two doses of CoronaVac followed by one dose of BNT162b2 (Pfizer-BioNTech) vaccine, 87 received two doses of ChAdOx1-S (AstraZeneca) followed by one dose of BNT162b2 (Pfizer-BioNTech) vaccine, and 32 received three doses of BNT162b2 (Pfizer-BioNTech) vaccine.

Considering immunogenicity data, within the 4-week follow-up period after the booster dose of vaccine, there were 22 incident cases of COVID-19 in patients with respiratory symptoms diagnosed more than 15 days after their third dose of SARS-CoV-2 vaccine: 12 cases (10.34%) with patients that received two doses of CoronaVac followed by one dose of BNT162b2 (Pfizer-BioNTech) vaccine, 8 cases (9.20%) with patients that received two doses of ChAdOx1-S (AstraZeneca) followed by one dose of BNT162b2 (Pfizer-BioNTech) vaccine, and 2 cases (6.25%) with patients that received three doses of BNT162b2 (Pfizer-BioNTech) vaccine. Four cases of COVID-19 were diagnosed 15 days after the first dose of vaccine. No difference in the rate of incident cases was found among the different vaccination schedules (Table 3).

Among patients diagnosed with COVID-19, the majority presented with mild symptoms of fatigue, weakness, changes in smell and taste, cough, and shortness of breath not meeting criteria for severity. Of the 28 infected patients, only 4 sought medical attention; all were seen at urgent care facilities and did not require hospitalization.

The analysis of the immunogenicity data showed an increase in seroconversion rate after the vaccine doses were administered, with no difference between vaccination schedules. Seropositivity was 39.47% at enrollment and 97.57% after the booster dose. Regarding IgG antibody titers (log-transformed), increases in geometric mean IgG titers were seen after each dose in the different vaccination schedules (log 1.86 at enrollment to log 7.06 after the third dose). Antibody titers after the first and the second dose varied between the different vaccine platforms: log 4.37 for CoronaVac, log 6.30 for ChAdOx-1, and log 7.09 for BNT162b2 (*p* < 0.001), with a statistically significant difference demonstrating the superiority of schedules containing ChAdOx-1 (AstraZeneca) and BNT162b2 (Pfizer-BioNTech) over CoronaVac (*p* < 0.001). However, after the third dose, IgG titers were similar across all vaccination schedules (*p* = 0.68) (Table 4).

An analysis excluding individuals with prior COVID-19 infection was conducted (N = 138). A seropositivity rate of 96% was observed after the booster dose. No statistically significant differences were found among different vaccine platforms after the first and second doses (*p* = 0.32). Regarding the geometric mean titer increase in IgG, the increase was independent of the vaccine platform, with a log increase from 0.49 at baseline to 6.72 after the third dose. However, the increase was lower for individuals who initially received the inactivated virus vaccine (CoronaVac) compared to those who received ChAdOx-1 (AstraZeneca) and BNT162b2 (Pfizer-BioNTech) after the second dose of the initial vaccine regimen, with log increases of 3.78 (1.93), 5.65 (2.06), and 6.60 (2.26), respectively (*p* < 0.001) (Figure 3).

Box plots show the distribution of log-transformed IgG titers over time. Y-axis: logarithm of IgG titers (Log_10_IgG titer); X-axis: vaccination schedule (combination of vaccines); boxes: represent the distribution of IgG titers for each vaccine group (mean and SD); lines (whiskers): minimum and maximum values, excluding outliers.

In the multivariate linear regression model, considering immunogenicity (IgG titer after the third dose of the vaccine) as the outcome variable, there is a tendency for both disease activity and the degree of immunosuppression to be associated with a reduction in IgG titers; however, none of the associations are statistically significant (*p* > 0.05%). Thus, there was no association between disease activity or degree of immunosuppression and anti-SARS-CoV-2 IgG titers after the third (booster) dose of vaccine in patients with SLE (*p* > 0.05%) (Table 5).

Considering the immunosuppressive drugs most frequently used in these patients, an analysis of seroconversion by specific immunosuppressive drug use at inclusion demonstrated that, after the second and third doses, individuals using mycophenolate had a significantly higher proportion of non-seroconversion (*p* = 0.005 and *p* = 0.011, respectively), with a seropositivity rate of 80% after the second dose and 91.8% after the third dose. There was no significant difference in the seroconversion rate between the groups using methotrexate or azathioprine (Table 6).

## 4. Discussion

The present study demonstrated high immunogenicity against SARS-CoV-2 in patients with SLE after a complete vaccination schedule (two doses + one booster dose), regardless of the vaccine platform administered. This is one of the first studies to evaluate the response to different immunization schemes—CoronaVac (inactivated SARS-CoV-2 vaccine), ChAdOx-1 (AstraZeneca), and BNT162b2 (Pfizer-BioNTech)—with an added booster dose in this patient population. Among the 235 patients with SLE included, an increase in antibody titers was observed after vaccination, with a seropositivity rate of 97.57% following a complete vaccination schedule, demonstrating a greater induction of humoral immunity when compared to a one- or two-dose homologous vaccination schedules. Furthermore, this prospective longitudinal study was also able to demonstrate the medium-term real-life effectiveness of COVID-19 vaccines in patients with an immune-mediated rheumatic disease.

Vaccination is a public health strategy to reduce mortality from infectious diseases at the population level. Before a vaccine can be recommended, its efficacy and effectiveness must be assessed; however, the measurement of these parameters in many population subgroups, including patients with immune-mediated rheumatic diseases, is severely limited by their exclusion from phase III trials [7]. Studies evaluating the incidence of hospitalization due to COVID-19 in patients with SLE demonstrated a risk approximately three times higher compared to the general population [18]. Our prospective cohort found 28 incident cases of SARS-CoV-2 infection, with a higher number of cases observed among patients who initially received CoronaVac, but without a statistically significant difference compared to other vaccine platforms; however, all were mild respiratory tract infections, with no hospitalizations or deaths, demonstrating a change from the pre-vaccination scenario in which a higher risk of unfavorable outcomes and mortality was observed in patients with SLE [19]. This was also reported in cohort studies comparing the outcomes of vaccinated and unvaccinated SLE patients relative to the general population [20]. Therefore, our findings corroborate the existing data on the protective effect of SARS-CoV-2 vaccines and provide further evidence of the importance of vaccination in patients with SLE. Regarding the still high number of cases, we point out that this increase evidenced after the third vaccine dose (booster dose) coincided with the period of circulation of the Delta variant in Brazil, a strain that is known to have greater infectivity compared to the original strain [21,22].

The immunogenicity of SARS-CoV-2 vaccines in patients with immune-mediated diseases has been the object of several studies since the first stages of development of the different COVID-19 vaccine platforms. However, the recommendation for the vaccination of patients with SLE was initially empirical [23]. A recent meta-analysis demonstrated a seropositivity rate of 81.1% in SLE patients who were vaccinated against COVID-19, a lower rate than that found in our study, in which seropositivity rose from 39.47% at enrollment to 97.57% after the completion of any vaccination schedule. This apparent superiority is mainly attributable to the studies included in the meta-analysis, which mostly evaluated outcomes after two doses of a SARS-CoV-2 vaccine; only two studies involving a booster-dose schedule were included [24].

On comparing the different vaccine platforms, we observed a smaller increase in IgG titers in patients who received the CoronaVac vaccine after the first and second doses. This finding is in line with previous studies of live inactivated SARS-CoV-2 vaccine platforms, in which a seroconversion rate of 70.4% was found in patients with immune-mediated diseases (versus 95.5% in the control group), as well as a lower increase in IgG titers [25].

The degree of immunosuppression in patients with SLE and their degree of disease activity are factors that may be related to the blunting of the vaccine response in these individuals. However, we found no such association after multivariate analysis. This is in contrast with data from previous studies in which immunosuppression was found to have an impact on the vaccine response, especially in patients receiving mycophenolic acid, glucocorticoids, and rituximab [26,27]. On analysis of these findings, we believe that the lack of difference in vaccine response in relation to the degree of immunosuppression is attributable to the fact that our study analyzed patients after they had received a booster dose, unlike previous studies that conducted outcome assessments after a two-dose schedule. The significant increase in humoral immunity in patients with an otherwise suboptimal response due to their degree of immunosuppression was also observed in another cohort that evaluated the effect of a booster dose in patients with immune-mediated disease who had received a homologous vaccination schedule with an inactivated virus or adenovirus vector vaccine [28]. On the other hand, when we analyzed seropositivity rates by the specific drug used at inclusion, we found that individuals who used mycophenolate had a significantly higher proportion of non-seroconversion. The results of this study suggest that the use of mycophenolate may be associated with a lower seroconversion rate, indicating a possible interference with the immune response, similar to findings previously described in the literature [29].

Our study has some limitations inherent to observational cohort designs. Although cases of COVID-19 infection were recorded before the vaccination period, some asymptomatic cases may have occurred during the intervals between vaccine doses, thus contributing to an increase in seropositivity in these patients. In an attempt to mitigate this effect, participants were contacted periodically by telephone so that they would not underestimate mild symptoms and thus fail to undergo confirmatory diagnostic testing.

Other important limitations of our study are related to the absence of a control group of a healthy population to compare the difference in the humoral response in patients with lupus in relation to the general population. Furthermore, our study only evaluated humoral immunity and did not extend to evaluating the impact on cellular immunity, which appears to play an important role in viral infections. Finally, as another limitation, we point out that this study only evaluated seropositivity based on IgG serology and not through neutralizing antibodies.

The strengths of this study are several, and include the number of participants enrolled, the length of follow-up, and the real-world setting; as it was conducted following the recommendations of health agencies during the pandemic, it provides an accurate picture of the response to a boosted vaccination schedule, as was recommended at the time for high-risk subgroups.

## 5. Conclusions

In conclusion, this study demonstrated the effectiveness of SARS-CoV-2 vaccines in patients with systemic lupus erythematosus receiving immunosuppressive therapy, confirming the importance of a complete (two-dose) vaccination schedule followed by a booster dose, which was associated with a significant increase in humoral immunity—especially for patients who received initial vaccination with a live inactivated vaccine.

## Figures and Tables

**Figure 1 vaccines-13-00127-f001:**
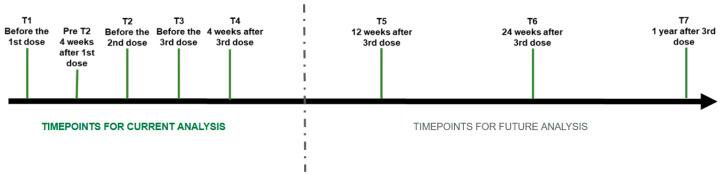
Flowchart of SAFER study, highlighting the different assessment timepoints. Vaccine platforms: CoronaVac, ChAdOx-1 (AstraZeneca), and BNT162b2 (Pfizer-BioNTech).

**Figure 2 vaccines-13-00127-f002:**
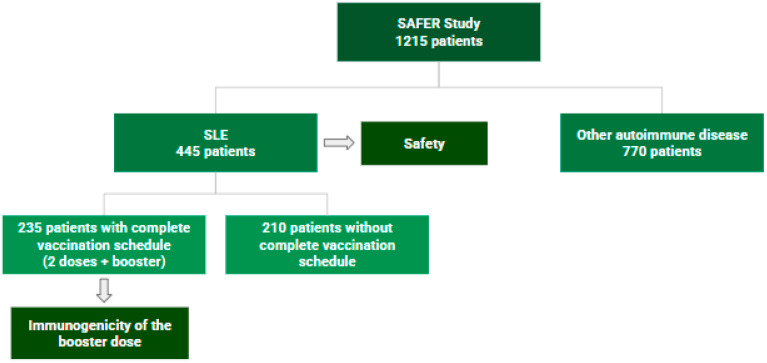
Organizational chart: patient inclusion and exclusion in study analysis.

**Figure 3 vaccines-13-00127-f003:**
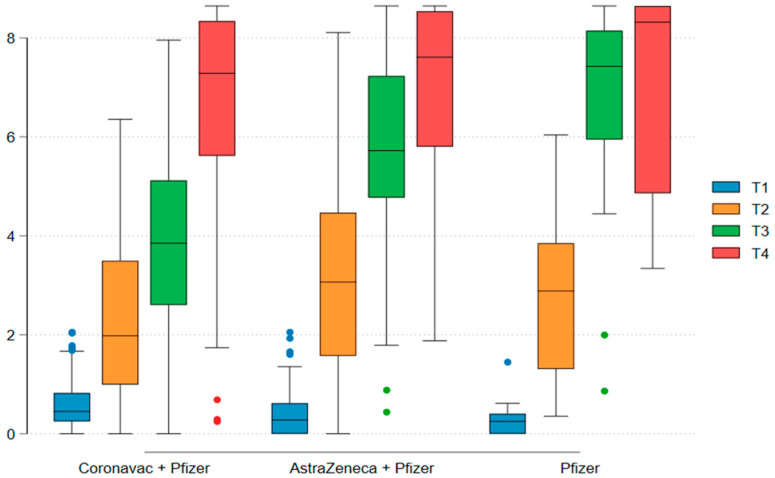
Comparison of IgG titers after different vaccination schemes, excluding pre-exposed group (n = 138).

**Table 1 vaccines-13-00127-t001:** Demographic and clinical characteristics of patients at inclusion.

	Total N = 235	CoronaVac + BNT162b2 N = 116	ChadOx-1+ BNT162b2 N = 87	BNT162b2 + BNT162b2 N = 32	*p*
**Sex**					0.540
Female	210 (89.36)	101 (87.07)	79 (90.80)	30 (93.75)	
Age, mean (SD)	38.0 (29.0–46.0)	35.5 (28.0–45.0)	40.0 (31.0–47.0)	37.5 (32.0–46.0)	0.095
Skin color					0.300
White	87 (37.02)	51 (43.97)	27 (31.03)	9 (28.13)	
Black	33 (14.04)	15 (12.93)	14 (16.09)	4 (12.50)	
Brown	109 (46.38)	46 (39.66)	45 (51.72)	18 (56.25)	
Disease in years, median (IQR)	10 (5–16)	8 (4–15)	12 (7–18)	7.7 (4–14)	0.006
Smoking	15 (6.38)	11 (9.48)	4 (4.60)	0 (0.00)	0.130
No comorbidities	92 (39.15)	59 (50.86)	23 (26.44)	10 (31.25)	0.001
Heart disease	10 (4.26)	5 (4.31)	2 (2.30)	3 (9.38)	0.220
Diabetes	11 (4.68)	6 (5.17)	4 (4.60)	1 (3.13)	1.000
Lung disease	7 (2.98)	1 (0.86)	2 (2.30)	4 (12.50)	0.007
Kidney disease	4 (1.70)	2 (1.72)	2 (2.30)	0 (0.00)	1.000
Hypertension	64 (27.23)	26 (22.41)	29 (33.33)	9 (28.13)	0.220
Obesity	27 (11.49)	12 (10.34)	10 (11.49)	5 (15.63)	0.710
Other comorbidities *	95 (40.43)	37 (31.90)	44 (50.57)	14 (43.75)	0.025
APS	19 (8.09)	11 (9.48)	6 (6.90)	2 (6.25)	0.800
Previous thrombosis	32 (13.62)	14 (12.07)	11 (12.64)	7 (21.88)	0.340
Disease activity					0.056
Remission	90/225 (40)	45/109 (41.28)	30/84 (35.71)	15/32 (46.88)	
Low activity	73/225 (32.44)	28/109 (25.69)	37/84 (44.05)	8/32 (25.00)	
Moderate-to-high activity	62/225 (27.56)	36/109 (33.03)	17/84 (20.24)	9/32 (28.13)	
Degree of Immunosuppression					0.160
No immunosuppression	70/232 (30.17)	40/115 (34.78)	25/86 (29.07)	5/31 (16.13)	
Low grade	27/232 (11.64)	9/115 (7.83)	12/86 (13.95)	6/31 (19.35)	
High grade	135/232 (58.19)	66/115 (57.39)	49/86 (56.98)	20/31 (64.52)	

Values are expressed as N (%) or N/N (%) for categorical variables and mean ± standard deviation (SD) or median (interquartile range [IQR]) for continuous variables; *p*-value < 0.05 was considered significant; APS, antiphospholipid syndrome; * hypothyroidism, osteonecrosis, dyslipidemia, osteoporosis, fibromyalgia.

**Table 2 vaccines-13-00127-t002:** Medications in use at inclusion according to the vaccination schedule.

	Total N = 235	CoronaVac + BNT162b2 N = 116	ChadOx-1 + BNT162b2 N = 87	BNT162b2 + BNT162b2 N = 32	*p*
Azathioprine	54/235 (22.98)	27/116 (23.28)	17/87 (19.54)	10/32 (31.25)	0.400
Oral corticosteroid	112/235 (47.66)	60/116 (51.72)	38/87 (43.68)	14/32 (43.75)	0.470
Oral corticosteroid dose					0.007
Up to 5 mg/day	54/112 (48.21)	24/60 (40.00)	25/38 (65.79)	5/14 (35.71)	
≥6 a 10 mg/day	27/112 (24.11)	14/60 (23.33)	7/38 (18.42)	6/14 (42.86)	
≥11 a 20 mg/day	19/112 (16.96)	10/60 (16.67)	6/38 (15.79)	3/14 (21.43)	
>20 mg/day	12/112 (10.71)	12/60 (20.00)	0/38 (0.00)	0/14 (0.00)	
Hydroxychloroquine	194/235 (82.55)	98/116 (84.48)	68/87 (78.16)	28/32 (87.50)	0.370
Mycophenolate	54/235 (22.98)	22/116 (18.97)	25/87 (28.74)	7/32 (21.88)	0.260
Methotrexate	32/235 (13.62)	12/116 (10.34)	14/87 (16.09)	6/32 (18.75)	0.330
Methotrexate dose					0.880
≤20 mg/week	23/32 (71.88)	8/12 (66.67)	10/14 (71.43)	5/6 (83.33)	
>20 mg/week	9/32 (28.13)	4/12 (33.33)	4/14 (28.57)	1/6 (16.67)	
Rituximab (regular use)	7/235 (2.98)	2/116 (1.72)	5/87 (5.75)	0/32 (0.00)	0.170

Values are expressed as N (%) or N/N (%); *p*-value < 0.05 was considered significant.

**Table 3 vaccines-13-00127-t003:** Number (%) of cases of SARS-CoV-2 infection after vaccination throughout follow-up.

COVID-19 Infection Cases	Total N = 235	CoronaVac + BNT162b2 N = 116	ChadOx-1 + BNT162b2 N = 87	BNT162b2 + BNT162b2 N = 32	*p*
Positive cases of SARS-CoV-2 more than 15 days after the 1st dose	4/235 (1.70)	3/116 (2.59)	1/87 (1.15)	0/32 (0.00)	0.800
Positive cases of SARS-CoV-2 more than 15 days after the 2nd dose	6/235 (2.55)	5/116 (4.31)	1/87 (1.15)	0/32 (0.00)	0.360
Positive cases of SARS-CoV-2 more than 15 days after the 3rd dose	22/235 (9.36)	12/116 (10.34)	8/87 (9.20)	2/32 (6.25)	0.860

Values are expressed as N (%) or N/N (%); *p*-value < 0.05 was considered significant.

**Table 4 vaccines-13-00127-t004:** Number (%) of seropositive patients and value of antibody titers according to different vaccine schedules and at different follow-up timepoints.

	Total N = 235	CoronaVac + BNT162b2 N = 116	ChadOx-1 + BNT162b2 N = 87	BNT162b2 + BNT162b2 N = 32	*p*
Serology					
Inclusion	90/228 (39.47)	47/114 (41.23)	30/84 (35.71)	13/30 (43.33)	0.660
28 days after the 1st dose	162/219 (73.97)	79/114 (69.30)	62/78 (79.49)	21/27 (77.78)	0.260
28 days after 2nd dose	200/219 (91.32)	98/110 (89.09)	74/79 (93.67)	28/30 (93.33)	0.500
After 3rd dose	201/206 (97.57)	99/103 (96.12)	73/74 (98.65)	29/29 (100.00)	0.370
IgG titer (Log10)					
Inclusion, mean (SD)	1.86 (1.89)	2.00 (1.88)	1.70 (1.90)	1.81 (1.94)	0.540
After 1st dose, mean (SD)	4.10 (2.49)	3.45 (2.04)	4.72 (2.71)	5.11 (2.82)	<0.001
After 2nd dose, mean (SD)	5.44 (2.17)	4.37 (1.80)	6.30 (1.98)	7.09 (1.86)	<0.001
After 3rd dose, mean (SD)	7.06 (1.80)	6.95 (1.95)	7.17 (1.65)	7.17 (1.59)	0.680

Values are expressed as N (%) or N/N (%) for categorical variables and mean ± standard deviation (SD) or median (interquartile range [IQR]) for continuous variables; *p*-value < 0.05 was considered significant. Seroconversion rates were evaluated in patients who underwent blood collection within a 30-day window (±7 days) post-vaccination. Participants with blood draws occurring outside this specified time period were excluded from the analysis.

**Table 5 vaccines-13-00127-t005:** Association between SLE activity or degree of immunosuppression and immunogenicity of COVID-19 vaccines, as assessed by anti-SARS-CoV2 antibody titers.

	Multivariate Linear Regression			
	Coefficient	CI [95%]		*p*
Disease activity				
Remission	-	-	-	-
Low activity	−0.503	−1.624	0.617	0.373
Moderate-to-high activity	−1.060	−2.134	0.013	0.053
Immunosuppression				
Without immunosuppression	-	-	-	-
Low degree of immunosuppression	−1.596	−3.744	0.551	0.143
High degree of immunosuppression	−0.692	−1.650	0.266	0.154

Coefficient represents the change in the outcome variable for an increase in the independent variable, holding other variables constant; CI, confidence interval; *p*-value < 0.05 was considered significant.

**Table 6 vaccines-13-00127-t006:** Seropositivity rate after vaccination in patients using different immunosuppressive drugs.

	Serology after 1st			
	Total N = 222	Seronegative N = 61	Seropositive N = 161	*p*
Mycophenolate	48 (21.62)	16 (16.23)	32 (19.88)	0.300
Methotrexate	31 (13.96)	10 (16.39)	21 (13.04)	0.520
Azathioprine	51 (22..97)	17 (27.87)	34 (21.12)	0.290
	Serology after 2nd			
	Total N = 222	Seronegative N = 21	Seropositive N = 201	*p*
Mycophenolate	51 (22.97)	10 (47.62)	41 (20.40)	0.005
Methotrexate	28 (12.61)	3 (14.29)	25 (12.44)	0.730
Azathioprine	49 (22.07)	4 (19.05)	45 (22.39)	1.000
	Serology after 3rd			
	Total N = 212	Seronegative N = 5	Seropositive N = 207	*p*
Mycophenolate	49 (23.11)	4 (80.00)	45 (21.74)	0.011
Methotrexate	29 (13.68)	0 (0.00)	29 (14.01)	1.000
Azathioprine	46 (21.70)	1 (20.00)	45 (21.74)	1.000

Values are expressed as (N) %; *p*-value < 0.05 was considered significant.

## Data Availability

The data are not publicly available due to privacy restrictions and are only available upon request to the corresponding author at nsartori@hcpa.edu.br.

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
