# Peer review of "Immunogenicity of SARS-CoV-2 Vaccination Schedules Including a Booster Dose in Patients with Systemic Lupus Erythematosus: Data from a Prospective Multicenter Study"

_vaccines, 2025, doi:10.3390/vaccines13020127_

Round 1
Reviewer 1 Report
Comments and Suggestions for Authors
The manuscript by Sartori et al. investigate the immunogenicity, in terms of antibody response, and the effectiveness to SARS-CoV-2 vaccination in patients with systemic lupus erythematosus in a multicentre study.
The authors show that SARS-CoV-2 vaccines are effective in reducing severe outcome and hospitalization. The booster dose induces a greater humoral response, especially in patients initially immunized with an inactivated virus vaccine schedule and those with a suboptimal humoral response.
There are some concerns to be addressed:
Introduction
Line 75: the authors refer to “several previous studies”, however they cite only one reference. Please, add more references.
Line 103: since the authors have evaluated only the humoral response, I would suggest to replace “immune response” with “humoral response”.
Methods
Line 114: Did the authors excluded patients who had COVID-19 prior to vaccination from the analyses? It is not clearly stated in the text. If yes, please add it in the exclusion criteria. If no, it is necessary to exclude them from the analyses because a previous SARS-CoV-2 infection represents a bias for the evaluation of both humoral and cellular response.
Line 138: Which is the cut-off used to define a positive response for Anti-Sars-CoV-2 S immunoassay? Please stated in the method’s section.
Results
Line 205: Replace “between” with “among” as more than 2 groups are compared.
How was the humoral response after vaccination in the SLE patients who had COVID-19? Have you investigated whether there is an association between the incidence of COVID-19 cases in SLE patients and the IgG titer, as well as disease activity or current therapy? Did patients who had COVID-19 show a lower antibody response or higher disease activity or were they on a particular therapy?
Did you collect blood samples also at the following time points after the third dose (T5, T6 and T7)? It would be interesting to evaluate in those patients who did not have COVID-19 during the follow up period, how long the antibody response persists.
Discussion
Line 273: The smaller increase in IgG titers in patients who received the CoronaVac vaccine may explain the higher number of COVID-19 cases reported in these subjects according to Table 4, although not significant. Please add a comment in the discussion.
A higher number of COVID-19 cases were reported after the thrid dose, after which a higher antibody response was observed. Did this period concurred with the Omicron wave in Brazil? It is known that the first-generation vaccines were designed against the Wuhan strain; thus, they offered reduced protection against the Omicron variant that is highly mutated in the spike protein compared to any others. Please add a comment on this.
The authors found 39.47% of seropositivity at the enrolment. As the subjects had not yet been vaccinated at this time, the observed seropositivity is probably due to the presence of cross-reactivity with other coronaviruses. Please add a comment on this.
Line 282: delete “on”
Are there data in the literature comparing the antibody response to SARS-CoV-2 vaccination in SLE patients versus healthy subjects? Please add this information in the discussion.
Limitations
- A major limitation of the study is the lack of a control group of healthy subjects. It would be important to compare the humoral response to vaccination in SLE patients with that of healthy subjects to understand how impaired their ability to induce an immune response to the COVID-19 vaccine is.
- Another limitation is the lack of assessment of the T-cell response, which, together with the antibody response, is another important component to consider when studying the immunogenicity of vaccines.
- The authors evaluate only the antibody response as IgG and not the neutralizing activity of these antibodies. Please add this limitation in the study.
Tables
Table 1: it is not specified the origin of the subjects. All the included patients are born in Brazil or some come from other countries? Please include this information in table 1.
Table 4: seroconversion rate are calculated on a lower number of patients. This is because the serological data were not available for all patients? Please specify the reason for the exclusion.
Did you exclude subjects who had COVID-19 from the analysis of the antibody response in the following time points reported in Table 4 and in graph 1?
Figures
- In figure 1 the line drawing is interrupted before T7.
- Refer to “Graph 1” as Figure 3 and specify in the y-axis what is shown (Log10IgG titer)
Author Response
"Please see the attachment."

Reviewer 2 Report
Comments and Suggestions for Authors
In their paper, Dr. Sartori et al provide interesting results on the proportion of seroreactivity and titres of anti-SARS-CoV2 antibodies following COVID-19 vaccination with three different vaccination schedules in a cohort of patients with SLE. I have added my comments into the pdf version of the manuscript.
My main objection of the study would be in regard to using the term (vaccine/vaccine schedule) effectiveness. With the study protocol not enrolling control subjects, I believe effectiveness can not be assessed and this terminology should be corrected in the absence of clear correlation between seropositivity and probability of protection against different clinical outcomes e.g. disease, hospitalisation, death.
I also think that study results would be more meaningful if the study period was selected more narrowly and in correlation with the timing of circulation of a particular SARS-CoV-2 strain, particularly so because the proportion of seropositive patients at enrolment was quite high (39.5%).
I also missed the information on the association between seropositivity rate or antibody titers and the probability of developing COVID-19 during the 12-month follow-up.

Round 2
Reviewer 1 Report
Comments and Suggestions for Authors
The authors answered to all my suggestions. The revised version of the article can be published.
Author Response
We are very grateful for the reviewer's contribution and time to our work.
Reviewer 2 Report
Comments and Suggestions for Authors
Please see the attachment.

Round 3
Reviewer 2 Report
Comments and Suggestions for Authors
I have no further comments.